# Development of 3D-Printed, Liquisolid and Directly Compressed Glimepiride Tablets, Loaded with Black Seed Oil Self-Nanoemulsifying Drug Delivery System: In Vitro and In Vivo Characterization

**DOI:** 10.3390/ph15010068

**Published:** 2022-01-05

**Authors:** Tarek A. Ahmed, Hanadi A. Alotaibi, Waleed S. Alharbi, Martin K. Safo, Khalid M. El-Say

**Affiliations:** 1Department of Pharmaceutics, Faculty of Pharmacy, King Abdulaziz University, Jeddah 21589, Saudi Arabia; halalotaibi@kau.edu.sa (H.A.A.); wsmalharbi@kau.edu.sa (W.S.A.); kelsay1@kau.edu.sa (K.M.E.-S.); 2Department of Medicinal Chemistry and the Institute for Structural Biology, Drug Discovery and Development, School of Pharmacy, Virginia Commonwealth University, Richmond, VA 23298, USA; msafo@vcu.edu

**Keywords:** glimepiride, black seed oil, SNEDDS, three-dimensional printed tablets, liquisolid technique, direct compression, pharmacokinetics

## Abstract

Glimepiride is characterized by an inconsistent dissolution and absorption profile due to its limited aqueous solubility. The aim of this study was to develop glimepiride tablets using three different manufacturing techniques, as well as to study their quality attributes and pharmacokinetics behavior. Black seed oil based self-nanoemulsifying drug delivery system (SNEDDS) formulation was developed and characterized. Glimepiride liquisolid and directly compressed tablets were prepared and their pre-compression and post-compression characteristics were evaluated. Semi-solid pastes loaded with SNEDDS were prepared and used to develop three-dimensional printing tablets utilizing the extrusion technique. In vivo comparative pharmacokinetics study was conducted on Male Wistar rats using a single dose one-period parallel design. The developed SNEDDS formulation showed a particle size of 45.607 ± 4.404 nm, and a glimepiride solubility of 25.002 ± 0.273 mg/mL. All the studied tablet formulations showed acceptable pre-compression and post-compression characteristics and a difference in their in vitro drug release behavior. The surface of the liquisolid and directly compressed tablets was smooth and non-porous, while the three-dimensional printing tablets showed a few porous surfaces. The inner structure of the liquisolid tablets showed some cracks and voids between the incorporated tablet ingredients while that of the three-dimensional printing tablets displayed some tortuosity and a gel porous-like structure. Most of the computed pharmacokinetic parameters improved with the liquisolid and three-dimensional printed tablets. The relative bioavailabilities of the three-dimensional printed and liquisolid tablets compared to commercial product were 121.68% and 113.86%, respectively. Therefore, the liquisolid and three-dimensional printed tablets are promising techniques for modifying glimepiride release and improving in vivo performance but more clinical investigations are required.

## 1. Introduction

Oral solid dosage forms, especially tablets, are the most popular among all administered medicines. Oral tablets are commonly prepared utilizing wet granulation, dry granulation or direct compression [1]. Recently, the freeze-drying technique has been utilized to develop oral tablets [2,3]. The most considerable drawbacks of these techniques are their inability to modify the dosage form to produce a complex multi-component system that meets individual needs in therapeutic regimens [4]. Alternatively, three-dimensional printing has been employed to develop dosage forms with single and multi-drug systems [4,5,6]. It is a manufacturing platform that holds a great promise in fabricating active pharmaceutical ingredients for personalized medicine. This concept could be achieved through tuning the printing process parameters, including the tablet’s size and layer thickness to control dosage, form and drug release. The use of three-dimensional printing technology in the development of personalized medicine and producing individualized drug dosages in the pharmaceutical industry could be useful [5]. Three-dimensional printing technology has become a powerful tool that has been employed in many pharmaceutical applications [7]. It is a novel drug formulation approach that can be used in the production of complex oral dosage delivery pharmaceuticals as tablets or pills using standard pharmaceutical materials in order to improve the drug release profile of the commercially available dosage forms [8].

Glimepiride, 1-(p-(2-(3-ethyle-4-methyl-2-oxo-3-pyrroline-1-carboxamido) ethyl) phenyl) sulfonyl)-3-(trans-4-methylcyclohexyl) urea, is a third generation antidiabetic sulfonylurea. The drug is used in the treatment of non-insulin dependent (type II) diabetes mellitus. The mechanism of glimepiride action involves stimulation of insulin release from beta cells in the pancreas, and by enhancing the activity of intracellular insulin receptors. Glimepiride is a relatively water insoluble drug (0.0384 mg/mL) with a pKa of 6.2. The drug undergoes slow gastrointestinal dissolution and exhibits inter-personal variations, which affects its bioavailability [9]. Severe hypoglycemia is a serious adverse effect that has been noted during glimepiride treatment with large doses due to variations in the absorption of the drug [10]. Different strategies have been reported to enhance the dissolution, oral absorption and bioavailability of glimepiride such as inclusion complexes, solid dispersion and micronized techniques [11,12,13,14]. Moreover, different glimepiride immediate and sustained release tablet formulations have been previously mentioned. Sublingual glimepiride tablets that are characterized by rapid drug absorption [15] and matrix type sustained release drug tablets that overcome the inconsistent drug dissolution and absorption profiles have been reported [10]. More investigations in this area using different techniques are required to achieve an oral dosage form characterized by enhanced drug bioavailability.

Medicinal plants have gained great interest from researchers and have been considered as alternatives or complementary agents to oral antidiabetic drugs because of their integrated effects [16]. Many studies have reported the traditional use of *Nigella sativa* (black seed) oil for the treatment of diabetes [17]. The low toxicity and side effects of this oil compared to oral medications are an extra advantage for its use [18]. *Nigella sativa* oil has been utilized to develop a self-nanoemulsifying drug delivery system (SNEDDS) of doxorubicin to enhance the anticancer activity against HepG2 cell lines [19]. Halder et al. reported the development of SNEDDS of black seed oil to improve the hypotriglyceridemic and hepatoprotective activities [20]. As far as we know, no black seed oil based-SNEDDS of glimepiride has been reported. To overcome glimepiride poor aqueous solubility, limited dissolution in the gastrointestinal fluids and to avoid dose dumping, we have developed a black seed oil based-SNEDDS formulation and used this formulation to prepare liquisolid and 3D-printed tablets. Directly compressed tablets containing pure glimepiride were also prepared and the in vitro and in vivo performances of the studied tablets were investigated. Black seed oil was incorporated in the SNEDDS formulation to improve glimepiride anti-diabetic activity.

## 2. Results and Discussion

SNEDDS is characterized by its ability to enhance the solubility, dissolution and absorption of many hydrophobic drugs in the gastrointestinal tract since it is self-emulsify quickly in the stomach aqueous contents and so introduces the loaded drug in solution within nano-sized oil droplets [21,22,23]. As a result of this behavior, the surface area and size of oil droplets that are easily digested are increased, and they are incorporated into mixed micelles that can pass through the intestinal lumen [24]. Additionally, SNEDDS offer a good alternative to the conventional oral formulations of lipophilic compounds [23]. Previously published works indicated the good stability and small globule size of SNEDDS prepared utilizing tween 80 and PEG 400 with different oils such as sefsol, linoleic acid, olive oil, oleic acid and isopropyl myristate [25].

Results for the prepared optimized SNEDDS formulation showed a particle size of 45.607 ± 4.404 nm, a polydispersity index of 0.389 ± 0.021 and a saturated glimepiride solubility of 25.002 ± 0.273 mg in each mL of the formulation. TEM image showed a discrete, spherical shape oil globule as illustrated in Figure 1. This droplet size was in proximity with that obtained using Malvern Zetasizer. The prepared SNEDDS formulation was used as a nano-carrier for glimepiride and utilized to develop oral tablets employing three different techniques.

### 2.1. Development of Glimepiride Liquisolid and Compressed Tablets

Pre-compression characteristics of the liquisolid and directly compressed tablets formulations powder blends were within the pharmacopeia acceptable range. The calculated angle of repose and Carr’s index for the liquisolid tablets formulation were 32.1° and 14.80%, respectively, while that of the directly compressed tablets were 26.2° and 10.66%, which is an indication of good flow. After compression of the two formulations, both types of tablets were subjected to quality control tests and the data obtained is illustrated in Table 1. The liquisolid and directly compressed tablets showed an average weight of 291.14 ± 2.54 and 181.683 ± 1.076 mg, respectively. The measured thickness was 4.067 ± 0.046 and 3.996 ± 0.026 mm for the liquisolid and directly compressed tablets, respectively. The diameter of both formulations was 9 mm since both were compressed using the same punch. Friability was less than 1%, which is within the acceptable range according to the specifications of the USP [26]. The liquisolid and directly compressed tablets showed a hardness of 97.01 ± 10.82 and 146.03 ± 9.17 N, respectively, indicating good mechanical strength.

### 2.2. Development of Three-Dimensional Printed Tablets

The two types of the prepared HPMC pastes formulations showed a pseudoplastic behavior with a value of flow index “n” less than 1. Non-SNEDDS gel (pure gel) paste and SNEDDS-based paste showed a viscosity values of 8604 ± 295 and 6648 ± 135 Pa·s, respectively, suggesting lower viscosity of the SNEDDS-based pastes formulations compared to the pure paste formulations. It is expected that incorporation of the SNEDDS formulation in the prepared paste lubricates the solid particles and prevent water loss. Zidan et al. reported similar behavior for carbopol based 3D-printing pastes [27].

After loading the prepared pastes into the syringe, the flow speed was adjusted at 2.6 mm/s. Lower flow speed resulted in incomplete deposition of the tablet layers while higher flow speed did not allow complete drying of the printed layer, resulting in poor construction of the tablet layer. Using a 0.58 mm printing nozzle allowed the free flowing of the pastes at the studied flow speed. Rahman et al. and Zhang et al. indicated the effect of the nozzle size on the flow index and mentioned that a narrow orifice resulted in slow flow, which resulted in an increase in the applied flow pressure while wide nozzle orifice (0.6 mm) enhanced the flow [28,29]. After drying of the tablets, marked reduction in the tablet weight was observed the effect that is attributed to the studied percent of the solid content in the developed paste. The prepared pastes were developed using 16% adsorbents (Neusilin, Avicel and FujiSil), 21% binder (PVPK90) and 8% disintegrant (Ac-Di-Sol). Pure three-dimensional printed tablet showed an average weight of 432.53 ± 21.76 mg while, tablets loaded with SNEDDS formulation showed an average weight of 551.68 ± 31.02 mg. A marked reduction in the tablet weight after drying has been previously reported [4,27]. The dried tablets demonstrated a thickness of 2.647 ± 0.160 and 3.239 ± 0.178 mm for the pure and SNEDDS-based formulation, respectively as represented in Table 1. The diameter of the dried tablets was 13.289 ± 0.485 and 14.855 ± 0.319 mm for the pure and SNEDDS-based tablets, respectively. Friability of the dried tablets was less than 1% which indicates good mechanical strength. The dried tablets showed a drug content of 1.92 ± 0.219–1.98 ± 0.201 mg.

It must be mentioned that the amount of Neusilin, Avicel and FujiSil (Adsorbents) and the percent of Ac-Di-Sol were fixed in all the prepared tablet formulations while, PVPK90 was added as a binder to the three-dimensional printed tablets due to the nature and method of development of this formulation.

### 2.3. In Vitro Release Study

The in vitro release profile of glimepiride from the prepared glimepiride tablets and commercial tablets is illustrated in Figure 2. The drug release characteristics of the prepared tablet formulations differed significantly. Glimepiride three-dimensional printed tablets showed an extended drug release behavior. This effect could be attributed to the presence of HPMC and PVPK90 in these tablets. Previous studies indicated that HPMC swells and forms a viscous gel like layer and a barrier upon hydration [4,30]. Moreover, incorporation of PVP of high molecular weight such as PVPK90 resulted in formation of controlled release system as previously mentioned by Kurakula and Rao [31]. Modulation of caffeine release from linear to bi-modal release pattern after incorporation of PVPK90 to HPMC matrix tablets was also mentioned by Ian et al. [32]. The drug release from the tablets loaded with SNEDDS was superior to that of the corresponding non-SNEDDS tablets and the commercial tablets. This effect could be attributed to the small size of the drug loaded SNEDDS, which provided large surface area for glimepiride release from the studied tablets.

The liquisolid tablets showed faster drug release behavior when compared to the three-dimensional printed tablets. Furthermore, the liquisolid tablets illustrated higher drug release than the directly compressed tablet. Similar behavior was also noticed with famotidine liquisolid tablets that demonstrated higher release than the directly compressed tablets [33]. Enhancement in the in vitro dissolution and drug release from the liquisolid formulation may be attributed to the formation of a liquisolid microenvironment with soft structures and high porosity, which favored the disintegration and dissolution process of drug loaded liquisolid formulation [34].

It is noteworthy that the 3D-printed tablets’ formulation was investigated in this work to shift from the conventional “one size fits all” formulation to personalized medicine as it is possible to change the drug concentration in the prepared formulation based on individual needs [35]. The studied excipients in the 3D-printed tablets achieved extended drug release behavior which add extra benefits to this formulation.

### 2.4. Scanning Electron Microscope (SEM)

SEM images for the surface of the three-dimensional printed tablet, either the SNEDDS-based or non-SNEDDS formulations, illustrated an irregular surface of some curvature with the presence of few pores. The surface revealed an appearance of tortuosity which may occur during fusion and solidification of the tablet layers (Figure 3). The surface of SNEDDS-based tablet formulation was more homogenous and less porous which may be due to the presence of the SNEDDS formulation that render the paste more consistent and decreased the rate of water evaporation during the drying step. Inner structure of the non-SNEDDS tablet showed a dried interlocking flakes like structure with some void spaces and pores. This behavior was previously reported for HPMC based sustained release matrix tablets [36]. SNEDDS based tablet illustrated a non-uniform gel like structure with a smaller number of void and pores which may be due to the incorporation of SNEDDS that makes the matrix more wet and decreased the rate of water loss. The surface of the liquisolid tablet and the directly compressed tablets demonstrated a smooth surface without any pores or channels while the inner structure of these tablets showed some cracks and voids between the incorporated tablet ingredients as previously mentioned for directly compressed matrix tablets [10]. Both liquisolid and direct compressed tablets showed dry surface and inner structure which could be attributed to the compression of the powder blend of both tablets using the same method. No drug crystals were observed in the liquisolid tablets which is an indication of complete solubilization of the drug in the SNEDDS formulation before it was adsorbed on the carrier.

### 2.5. In Vivo Study

A comparative pharmacokinetics study was conducted to investigate the in vivo behavior of the prepared tablet formulations. The obtained data for the plasma glimepiride level versus time were used to construct the drug profile and to calculate the pharmacokinetic parameters. Figure 4 illustrates the plasma concentration versus time profile of glimepiride following the oral administration of the prepared tablets to the studied animals.

Table 2 demonstrates the calculated pharmacokinetics parameters. The liquisolid and three-dimensional printed tablets showed an improvement in most of the calculated pharmacokinetic parameters when compared to the directly compressed tablets and the commercial tablets. This behavior could be attributed to the existence of the drug in the nano-sized form which gives rise to better absorption. The liquisolid and three-dimensional printed tablets showed almost the same maximum plasma concentration over time, time to reach maximum drug concentration, absorption and elimination rate constants, area under the curve, total clearance rate, apparent volume of distribution and mean residence time. Although these formulations (liquisolid and three-dimensional printed tablets) were prepared using two different techniques and they demonstrated a marked difference in the in vitro release profile, administration of all the studied formulations as a suspension diminished the benefits of the liquisolid technique. This finding was previously mentioned for lyophilized orodispersible tablets [37] and fast dissolving tablets [38]. A clinical study on human volunteers is recommended to investigate the difference in disintegration and release characteristics from intact tablets. The relative bioavailabilities of the three-dimensional printed and liquisolid tablets compared to directly compressed tablets were 187.36% and 181.68%, respectively. Additionally, the relative bioavailabilities of these tablets compared to the marketed product were 121.68% and 113.86%, respectively. The obtained results indicate that SNEDDS-based tablets developed using the liquisolid and three-dimensional printing have a positive impact on the rate and extent of glimepiride absorption, the effect that could be attributed to the presence of the drug in the nano-sized form as previously mentioned. As a result, it could be concluded that improving the drug solubility and dissolution is expected to improve the bioavailability of poorly soluble drugs and those with a slow dissolution rate, such glimepiride.

Glimepiride commercial tablets “Amaryl^®^ of Hoechst Marion Roussel, Stockholm, Sweden” showed a time point to reach the maximum plasma concentration of 2 h, a maximum plasma concentration of 2058.667 ± 128.811 ng/mL, an elimination half-life of 7.247 ± 1.095 h, an area under the plasma level time curve from zero time to the last measurable concentration of 14,021.671 ± 1539.859 ng/mL·h, and a mean residence time of 10.139 ± 0.701 h, as previously revealed in our most recent publication [35]. These results also indicate superiority of SNEDDS-based tablets developed using the liquisolid and three-dimensional printing.

## 3. Materials and Methods

### 3.1. Materials

Glimepiride was a kind gift by the Saudi Pharmaceutical Industries & Medical Appliances Corporation (SPIMACO) (Alqasim, Saudi Arabia). Croscarmellose sodium (Ac-di-sol) was obtained from Biosynth International, Inc. (San Diego, CA, USA). Microcrystalline cellulose (Avicel) PH-101 was purchased from Winlab laboratory chemicals (Leicestershire, UK). Hydroxypropyl methyl cellulose (HPMC) 4000 cp was procured from Spectrum Chemical Manufacturing Corporation (Gardena, CA, USA). Lactose anhydrous, polyethylene glycol (PEG) 400, tween 80, fumed-silica (0.007 mm), polyvinyl pyrrolidone (PVP) with a molecular weight of 360,000 Da (K90), Methocel^®^ A15 LV, 27.5–31.5% methoxyl basis were all obtained from Sigma-Aldrich Inc. (St. Loius, MO, USA). Talc powder was procured from Whittaker Clark & Daniels (South Plainfield, NJ, USA). Magnesium stearate was obtained from Winlab Laboratory Chemicals Reagents (Leicestershire, UK). Neusilin (Magnesium aluminometasilicate) and FujiSil (porous silicon dioxide) were obtained as a kind gift from Fuji Chemical Industry Co. Ltd. (Tokyo, Japan).

### 3.2. Preparation and Characterization of Glimepiride Loaded SNEDDS

In this study different oils, surfactants and cosurfactants were screened to select the three components utilized to develop SNEDDS. The solubility of glimepiride was studied in five different oils with antidiabetic activity namely; avocado oil, black seed oil, cactus pear seed oil, coconut oil and flaxseed oil. Glimepiride showed the highest solubility in black seed oil, tween 80 and PEG 400 (Data not shown). Mixture design of extreme vertices was utilized to develop an optimized SNEDDS formulation of small particle size. The studied variables were: black seed oil, tween 80 and PEG 400 in the range of 10–40%, 10–40% and 40–80%, respectively. An optimized SNEDDS formulation that contains 10%, 40% and 50% of black seed oil, between 80 and PEG 400, respectively, was proposed. This SNEDDS formulation was prepared and characterized as described below. Briefly, 1 g of SNEDDS formulation was prepared in a screw cap vials by accurately weighing 100, 400 and 500 mg of black seed oil, surfactant and cosurfactant, respectively. The mixture was vortex until a homogenous dispersion of the three components was obtained.

The known weight of the SNEDDS formulation was added to a specified volume of distilled water, in a ratio of 1:10 (*w*/*v*), on a magnetic stirrer until a homogenous yellowish nanoemulsion was obtained. Malvern Zetasizer Nano ZSP, Malvern Panalytical Ltd. (Malvern, UK) was used to evaluate the size and polydispersity index of the prepared nanoemulsion. The average of three readings was recorded.

Few drops of the prepared SNEDDS formulation were mounted on a carbon-coated grid. The sample was left for about 2 min before examination using transmission electron microscopy (TEM) model JEM-1230 (JOEL, Tokyo, Japan).

Drug solubility in the prepared SNEDDS formulation was evaluated by adding excess amount of glimepiride to a screw-capped vial containing 3 mL of SNEDDS. The vial was placed in a thermostatically controlled shaking water bath (Model 1031; GFL Corporation, Burgwedel, Germany) at 25 ± 0.5 °C. After 72 h, the vial was taken and the mixture was filtered through a 0.22-mm pore-size Millipore filter. Glimepiride concentration was determined spectrophotometrically at 230 nm. The experiment was performed in triplicate.

### 3.3. Preparation and Characterization of Glimepiride Liquisolid Tablets

Glimepiride liquisolid tablets were prepared using Neusilin and avicel PH-101 as a carrier. FujiSil was used as a coating material. Ac-Di-Sol was added as a disintegrant. Briefly, glimepiride (2 mg) was dissolved in 100 mg of the prepared SNEDDS and triturated well with 150 mg of the carrier (149.53 mg Neusilin and 0.47 mg avicel) and 10 mg of FujiSil. The mixture was de-lumped through a mesh sieve (no. 40) and mixed with 20.8 mg of Ac-Di-Sol (8% *w*/*w* based on the total weight of the carrier, coating material and SNEDDS) for 15 min. The glidant (talc 0.5% *w*/*w*) and lubricant (magnesium stearate 0.5% *w*/*w*) were also de- lumped through the same mesh sieve, then added to the powder blend and mixed for 3 min.

Flow properties of the prepared powder mixture were estimated by calculating the angle of repose following the specification stated in the powder flow chapter of the USP29-NF24 (General Chapters: 1174) [39]. Furthermore, compressibility of the powder was evaluated by determining the bulk and tap densities [39].

A single punch Erweka tablet press machine equipped with 9 mm flat face round punches (Heusenstamm, Germany) was used to develop the liquisolid tablets from the prepared powder blend. The post-compression characteristics, including weight, thickness, diameter, friability and drug content were tested according to the specifications mentioned in the United States Pharmacopeia for quality control tests of tablets [40].

Pure drug loaded directly compressed tablets were also prepared using the same ingredients mentioned above. The prepared tablets were evaluated for pre- and post-compression characteristics as described above.

### 3.4. Development of Three-Dimensional Printed Tablets

HPMC gel formulations with and without SNEDDS formulation were studied. Non-SNEDDS gel (pure gel) was prepared by dispersing known weight of glimepiride in 20 mL of distilled water over a magnetic stirrer. HPMC (4% *w*/*v*) was gradually added to the mixture while stirring. The obtained medicated polymeric mixture formulation was left overnight at 4 °C in a refrigerator to allow complete swelling of the polymer particles and formation of viscous gels. For SNEDDS-based gels, 2 g of glimepiride loaded SNEDDS formulation was added to 18 mL of distilled water over a magnetic stirrer. HPMC was gradually added to the mixture with continuous stirred. The medicated polymeric mixture was left overnight in the refrigerator.

Mixture of Neusilin and Avicel (15% *w*/*w*), and FujiSil (1% *w*/*w*) were used as insoluble ingredient to facilitate adsorption of the SNEDDS formulation. PVPK90 (21% *w*/*w*) was added as a binder. Ac-Di-Sol (8% *w*/*w* based on the total weight of adsorbent, binder and SNEDDS) was used as a disintegrant. The dried powders were well-blended and transferred to a mortar containing the prepared HPMC gel matrix (20 g). The studied ingredients were continuously mixed until a smooth homogenous paste was obtained.

Viscosity values of the prepared pure and SNEDDS-loaded pastes were estimated using Kinexus oscillation rheometer (Malvern Instruments Ltd., Worchestershire, UK) utilizing the rSpace (version 2.0) package software.

The medicated paste formulations were developed into three-dimensional printed tablets utilizing a REGEMAT3D V1 BioPrinter (REGEMAT Inc., Granada, Spain) using a computer-aided design (CAD) modelling software (REGEMAT 1.4.9 Designer, Granada, Spain). Each paste was loaded into a syringe and extruded through a 0.58 mm printing nozzle with a flow speed of 2.6 mm/s and infill speed of 10 mm/s. A total of eight layers were printed layer-by-layer until a cylindrical tablet of 15 mm diameter was obtained. A patch of 20 tablets was prepared using 20 g of the gel matrix. The printed tablets were kept in a vacuum dryer at 40 °C for 24 h to dry and stored in a hermetically sealed container. Quality control tests of the prepared tablets were assessed as mentioned above.

### 3.5. In Vitro Release Study

The in vitro release of glimepiride from the liquisolid, directly compressed, three-dimensional printed tablets and commercially available drug product “Amaryl^®^ of Hoechst Marion Roussel, Stockholm, Sweden” was studied using Erweka GmbH DT 700, paddle type (type II) USP dissolution test apparatus, DT 700 LH device (Heusenstamm, Germany). The experiment was carried out in 900 mL of distilled water containing 0.1% sodium lauryl sulfate (SLS) at 37 °C. SLS was added as a commonly used surfactant in dissolution media for poorly water soluble drugs [41]. The paddle speed was adjusted at 75 rpm. Samples of 5 mL, with immediate replacement, were withdrawn at 0.25, 0.5, 1, 2, 3, 4, 5, 6 and 12 h. The collected samples were filtered and assayed spectrophotometrically at 230 nm for glimepiride content. The profile for glimepiride release were constructed. The experiment was conducted in triplicate.

### 3.6. Scanning Electron Microscope (SEM)

To study the morphology of the prepared tablet formulations, SEM images were captured. Samples from the prepared tablet surface and inner layers were prepared, using a surgical scalpel, and mounted onto aluminum stubs and sputter-coated with gold. Images of the inner and surface structure were taken at accelerating voltage of 10 kV using Philips XL30 SEM (Eindhoven, The Netherlands) to study the.

### 3.7. In Vivo Study

The pharmacokinetics of glimepiride from the prepared liquisolid, three-dimensional printed and directly compressed tablets were studied utilizing a single glimepiride dose one-period parallel design. Male Wistar rats weighing 200–250 g were used. The animals were divided into three groups, with six animals in each group. Group I was administered the directly compressed tablets formulation. Group II received liquisolid tablets formulation. Group III received the three-dimensional printed tablet formulation. Before animal treatment, the rats were maintained in a temperature-controlled closed area with a 12 h light/dark cycle for one week. Animals were kept with free access to food and water. A glimepiride dose of 10 mg/kg was used [42]. Each tablet formulation was crushed and suspended in 1% carboxymethyl cellulose to facilitate its administration to the animals through a gastric tube as previously mentioned with simvastatin liquisolid tablets [43] and rosuvastatin orodispersible tablets [37]. Blood samples (250 µL) were collected from the animals through retro-orbital puncturing under light ether anesthesia at 0.5, 1, 2, 4, 6, 8, 12 and 24 h. The collected plasma samples were immediately separated by centrifugation at 6000 rpm for 10 min and stored at −20 °C. The protocol for this study received prior approval from the Research Ethics Committee, Faculty of Pharmacy, King Abdulaziz University, KSA (Reference No. 1021442). The study was performed following the guidelines of Good Clinical Practice, the International Conference on Harmonization, and the European Medicines Agency [44,45].

For determination of glimepiride in the plasma samples, calibration standards were prepared from glimepiride and metformin (internal standard) methanolic stock solutions. Glimepiride plasma concentration in the unknown and the prepared calibration standards was determined using Perkin Elmer high-performance liquid chromatograph (HPLC) equipped with variable wavelength UV detector and auto sampler. Separation was done using Phenomenex, RP Hi-Q-Sil C18 column (250 × 4.6 mm, 5 µm, Phenomenex Inc., Torrance, CA, USA). The mobile phase comprised of 64% acetonitrile and 36% potassium dihydrogen orthophosphate (0.02 M) adjusted to pH 3.5. The flow rate of the mobile phase was adjusted at 1 mL/min. The UV detector was adjusted at 230 nm. For preparation and extraction of the samples; 1 mL of acetonitrile-methanol (1:1) mixture was added to each plasma sample and the mixture was subjected to vortex for 1 min followed by centrifugation for 10 min at 5000 rpm. The organic phase was separated and completely evaporated to dryness under constant stream of nitrogen at 50 °C. The residue was reconstituted in 80 µL of the mobile phase and an injection volume of 30 µL was used. The HPLC method utilized for quantification of glimepiride in the plasma samples was reproduced except for slight modification [46].

A non-compartmental pharmacokinetic model utilizing Kinetica^TM^ software (version 4, Thermo Electron Corporation, Waltham, MA, USA) was used to calculate the pharmacokinetics parameters. The drug plasma concentration versus time was constructed and the maximum plasma concentration over time and the time point to reach the maximum plasma concentration were specified. The area under the drug plasma concentration- time curve from time zero to last measurable concentration (AUC0–24) was determined using the linear trapezoidal method. The area under the plasma concentration-time curve from time zero to infinity (AUC0–end) was calculated as the sum of the (AUC0–24) plus the ratio of the last measurable plasma concentration to the elimination rate constant. The area under the first moment of the plasma concentration-time curve was determined from the area under the concentration times versus the time curve. The mean residence time (MRT0–inf) was calculated from the ratio of AUMC to AUC. The elimination half-life was calculated as 0.693/elimination rate constant. The total body clearance was calculated by dividing the dose by AUC. The apparent volume of distribution was calculated by multiplying the total body clearance by mean residence time. Finally, the relative bioavailability of the studied tablets was determined using AUC test/AUC reference × 100.

## 4. Conclusions

SNEDDS loaded glimepiride tablets were successfully developed using the liquisolid and three-dimensional printing techniques. The liquisolid tablets showed fast in vitro drug release while the three-dimensional tablets demonstrated a controlled drug release behavior. Marked differences in the surface and inner structure of the prepared tablets were noticed. The liquisolid and three-dimensional printed tablets showed an improvement in most of the calculated pharmacokinetic parameters when compared to the directly compressed tablets and the commercial tablets. Furthermore, both tablet formulations illustrated higher relative bioavailabilities when compared to either the directly compressed tablets or the commercially available drug product. Therefore, the prepared liquisolid and three-dimensional printed tablet formulations are promising glimepiride solid dosage forms but clinical study is recommended.

## Figures and Tables

**Figure 1 pharmaceuticals-15-00068-f001:**
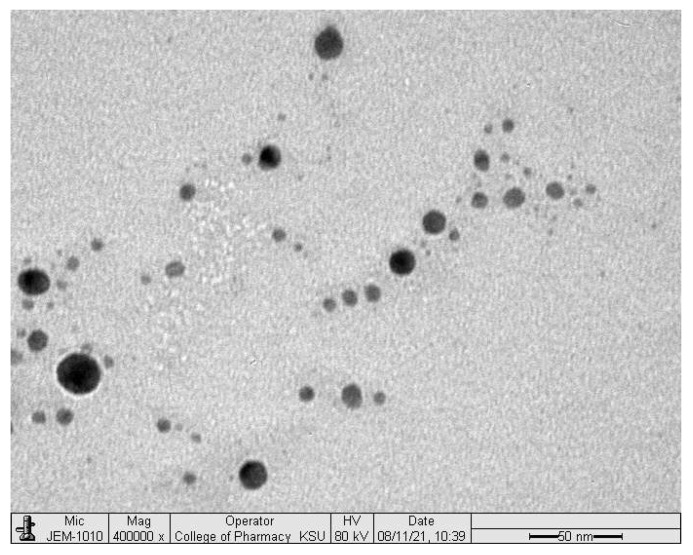
Transmission electron microscope image of the prepared SNEDDS formulation.

**Figure 2 pharmaceuticals-15-00068-f002:**
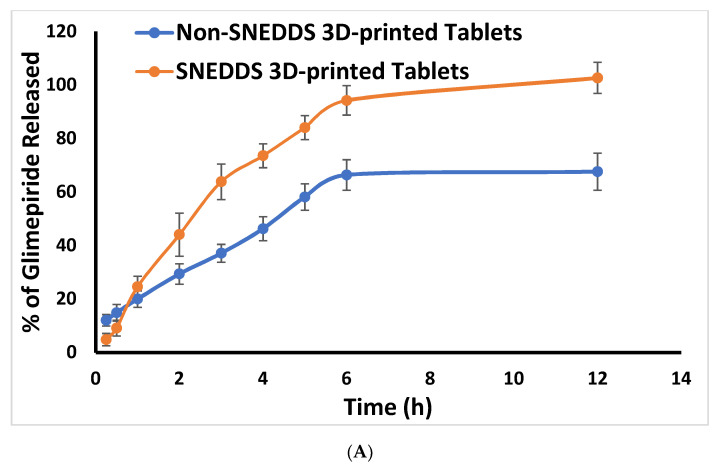
In vitro release of glimepiride from; (**A**) the prepared 3D-printed tablets, (**B**) the liquisolid, direct compressed and commercial tablets.

**Figure 3 pharmaceuticals-15-00068-f003:**
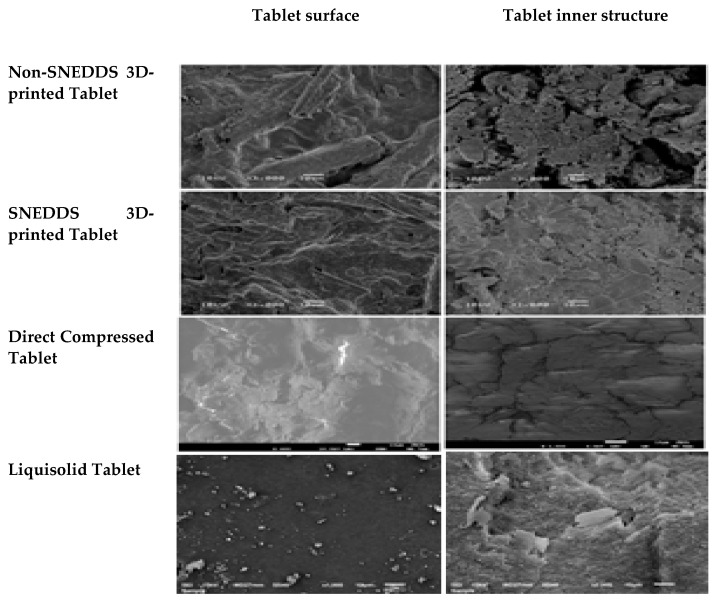
SEM images for the surface and inner structure of the prepared tablets.

**Figure 4 pharmaceuticals-15-00068-f004:**
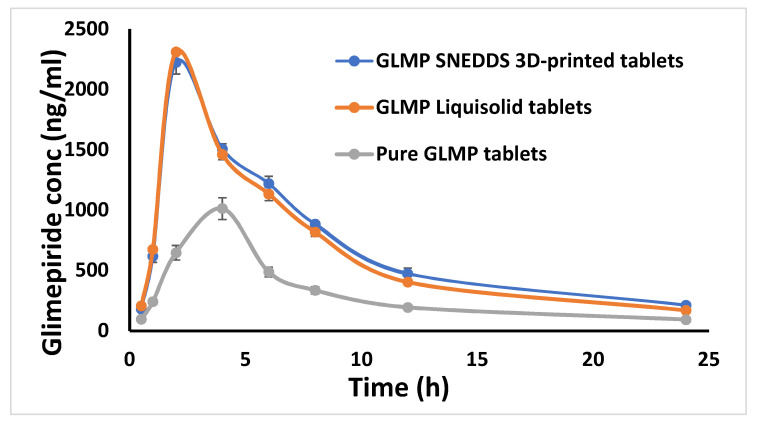
The plasma concentration versus time profile of glimepiride following the oral administration of the prepared tablets to Male Wistar rats. (Mean ± SD, *n* = 6).

**Table 1 pharmaceuticals-15-00068-t001:** Composition and quality control tests of the prepared glimepiride tablets.

	Parameter	Three-Dimensional Printed Tablets		Liquisolid Tablets	Directly Compressed Tablets
		Non-SNEDDS	SNEDDS		
Composition	Drug carrier/vehicle	Distilled water	SNEDDS	SNEDDS	-
	Carrier (mg)	-	100	100	-
	Neusilin (mg)	149.53	149.53	149.53	149.53
	Avicel (mg)	0.47	0.47	0.47	0.47
	FujiSil (mg)	10	10	10	10
	PVPK90	210	210	-	-
	Ac-Di-Sol (mg)	21.6	37.6	20.8	12.8
Characteristics	Actual Weight (mg)	432.53 ± 21.76	551.68 ± 31.02	291.14 ± 2.54	181.683 ± 1.076
	Thickness (mm)	2.647 ± 0.160	3.239 ± 0.178	4.067 ± 0.046	3.996 ± 0.026
	Diameter (mm)	13.289 ± 0.485	14.855 ± 0.319	9 ± 0	9 ± 0
	Friability (%)	0.159	0.117	0.019	0.384
	Drug content (mg)	1.98 ± 0.201	1.92 ± 0.219	2.080 ± 0.058	1.89 ± 0.46
	Disintegration time (min)	>60	>60	3.59 ± 0.68	8.40 ± 0.05

Abbreviations: SNEDDS, self-nanoemulsifying drug delivery system. Notes: Talc (3.01 mg) and Magnesium stearate (3.01 mg) were added to the liquisolid and directly compressed tablets as glidant and lubricant, respectively. HPMC (4%) was used in the three-dimensional printed tablets as a gel matrix. Three-dimensional printed tablets showed long disintegration time “more than 60 min” which is a common characteristic of the extended-release tablet.

**Table 2 pharmaceuticals-15-00068-t002:** Pharmacokinetic parameters of glimepiride following oral administration of the studied tablets to Male Wistar rats (*n* = 6).

Pharmacokinetic Parameters	Three-Dimensional Printed Tablets	Liquisolid Tablets	Directly Compressed Tablets
T_max_ [h]	2 ± 0	2 ± 0	4 ± 0
C_max_ [ng/mL]	2223.671 ± 96.877	2310.333 ± 20.793	1013.333 ± 7.214
K_ab_ [h^−1^]	0.894 ± 0.004	0.9021 ± 0.0008	1.547055 ± 0.029302
t_½ab_ [h]	0.776 ± 0.003	0.7680 ± 0.0007	0.448056 ± 0.008503
K_el_ [h^−1^]	0.098 ± 0.004	0.1071 ± 0.0015	0.086805 ± 0.007111
t_½el_ [h]	7.083 ± 0.272	6.456 ± 0.089	8.0195 ± 0.6672
V_d_ [(mg)/(ng/mL)]	0.0023 ± 0.0008	0.0023 ± 0.0001	0.00614 ± 0.00031
TCR [ml/min]	0.0037 ± 0.0004	0.0041 ± 0.0001	0.0089 ± 0.0003
AUC_(0–24)_ [ng/mL × h]	17,061.58 ± 469.14	15,965.751 ± 260.806	7330.917 ± 170.392
AUC_(24–end)_ [ng/mL × h]	1573.996 ± 1360.753	2105.403 ± 29.163	2615.314 ± 217.465
AUC_(0–end)_ [ng/mL × h]	18,635.582 ± 1157.246	18,071.153 ± 246.925	9946.230 ± 362.789
AUMC_(0–24)_ [ng/mL × h^2^]	131,177.601 ± 3105.725	115,564.667 ± 2328.201	56,315.921 ± 2452.962
AUMC_(24–end)_ [ng/mL × h^2^]	38,034.13 ± 32,210.89	50,529.667 ± 699.902	62,767.530 ± 5219.163
AUMC_(0–end)_ [ng/mL × h^2^]	169,211.711 ± 31,109.462	166,094.334 ± 2230.636	119,083.413 ± 7549.560
MRT [h]	9.035 ± 1.175	9.191 ± 0.056	11.965 ± 0.339

Abbreviations: T_max_, the time point to reach the maximum plasma concentration; C_max_, the maximum plasma concentration over the time specified; K_ab_, absorption rate constant; K_el_, elimination rate constant; t½_ab_, absorption half-life; t½_el_, elimination half-life; TCR, total clearance rate; AUC, area under the plasma concentration time curve; AUMC, the area under the first moment of the plasma concentration-time curve; MRT, the mean residence time; V_d_, the apparent volume of distribution.

## Data Availability

Data is contained within the article.

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
