# Peer review of "Development of 3D-Printed, Liquisolid and Directly Compressed Glimepiride Tablets, Loaded with Black Seed Oil Self-Nanoemulsifying Drug Delivery System: In Vitro and In Vivo Characterization"

_pharmaceuticals, 2022, doi:10.3390/ph15010068_

Round 1
Reviewer 1 Report
The authors have made an effort to properly address the comments of the reviewers in the revised manuscript.
The manuscript is much improved and can be published in Pharmaceuticals in the present form.
Author Response
Many thanks.
Reviewer 2 Report
- The change of the title, necessarily forces the objective of the study to change. "The aim was to study the pharmacokinetics behavior of glimepiride tablets developed using three different techniques", now, the objective cannot be limited to pharmacokinetics behavior, since the development of 3D-printed, liquisolid and directly compressed glimepiride tablets, necessarily implies other experimental trials that of course the authors incorporated into their manuscript.
- Summary: Verify the units of solubility ... 25 ± 0.273 mg / ?, also report the decimal figures of solubility which should be equal to the uncertainty 25.000 ± 0.273, See too, the value of particle size (45.600 ± 4.404 nm)
- review all the quantities and their respective uncertainties, in most cases the decimals of the value do not coincide with the number of decimal places of the uncertainty. Example: 45.6 ± 4.404 nm; 25 ± 0.273, 181.68± 1.076 (See table 2; 0.086805±0.0071-->0.0868±0.0071)
Author Response
1. The change of the title, necessarily forces the objective of the study to change. "The aim was to study the pharmacokinetics behavior of glimepiride tablets developed using three different techniques", now, the objective cannot be limited to pharmacokinetics behavior, since the development of 3D-printed, liquisolid and directly compressed glimepiride tablets, necessarily implies other experimental trials that of course the authors incorporated into their manuscript.
Reply
The aim has been modified in the revised manuscript.
2. Summary: Verify the units of solubility ... 25 ± 0.273 mg / ?, also report the decimal figures of solubility which should be equal to the uncertainty 25.000 ± 0.273, See too, the value of particle size (45.600 ± 4.404 nm)
Reply
The required information has been added to the revised manuscript.
3. review all the quantities and their respective uncertainties, in most cases the decimals of the value do not coincide with the number of decimal places of the uncertainty. Example: 45.6 ± 4.404 nm; 25 ± 0.273, 181.68± 1.076 (See table 2; 0.086805±0.0071-->0.0868±0.0071)
Reply
All the quantities and their respective uncertainties have been reviewed and modified.
This manuscript is a resubmission of an earlier submission. The following is a list of the peer review reports and author responses from that submission.
Round 1
Reviewer 1 Report
The objective of the research was fulfilled, the authors present in a detailed and precise way the pharmacokinetic behavior of three tablets loaded with glimepiride.
The authors present a good research work that can be published in its current form, however, I advise to greatly improve the graphs. These images can be edited in specialized software such as LaTEX.
Author Response
Kindly find the attached file for point-by-point response to the reviewer’s comments.

Reviewer 2 Report
General comments
The subject of the study falls within the scope of the Pharmaceuticals since it describes the preparation and characterization of liquisolid, directly compressed and 3D-printed glimepiride tablets, as well as their pharmacokinetic behavior evaluation.
However, the paper needs significant corrections, clarifications, and supplementation, as described below in the specific comments to the authors.
Specific Comments:
- Title: The title should emphasize on the tablet preparation and characterization and not on the short and questionable part (see below comments) of the PK study (e.g., Development of 3D-printed, liquisolid and directly compressed glimepiride tablets, loaded with black seed oil self-nanoemulsifying drug delivery system: in vitro and in vivo characterization)
- Abstract: It should be corrected/re-written after corrections/clarifications and supplementation according to the comments below. Furthermore, the animal model used in the PK study should be specified in the abstract.
- Introduction: Glimepiride tablets are commercially available and in the SPC of the products is stated that the drug is rapidly and completely absorbed (100%) from the intestine after oral administration. The authors do not refer to the characteristics commercially available formulations in the introduction part, but they point that “the drug undergoes slow gastrointestinal dissolution and exhibits inter-personal variations, which affects its bioavailability”, giving only one old reference (ref 9: Badian et al, 1994). The authors should document their statement based on recent bibliography. In addition, the authors state that “Different strategies have been reported to enhance the dissolution and oral absorption of glimepiride, especially from tablets, such as solid dispersion and micronized techniques” giving only one relatively old research paper as reference (ref 11: Xiao Ning et al., 2011). However, a short search in literature revealed several papers dealing with the development of immediate or sustained release glimepiride tablet formulations (e.g., Qushawy et al., Sci. Pharm. 2020, 88, 52, https://doi.org/10.3390/scipharm88040052; Dhanorkar and Shah, Future J Pharm Sci (2021) 7:225, https://doi.org/10.1186/s43094-021-00374-5; Jogala et al., 2016, Int J Pharm Pharm Sci, Vol 8, Issue 5, 271-278; Gill et al., Asian Journal of Pharmaceutics - July-September 2010, http://doi.org/10.4103/0973-8398.72121; Glob J Pharmaceu Sci 1(2): Pagadala et al., GJPPS.MS.ID.55553, 2016; Siwach, et al., Nanoscience & Nanotechnology-Asia 10(5), 2020, http://doi.org/10.2174/2210681209666190422160115; Li et al., International Journal of Nanomedicine 2016:11 3777–3788, http://dx.doi.org/10.2147/IJN.S105419; Al-Madhagi et al., Journal of Pharmaceutics Volume 2017, Article ID 3690473, 5 pages https://doi.org/10.1155/2017/3690473; Arpita Pal et al., ACS Omega 2020, 5, 19968−19977, http://pubs.acs.org/journal/acsodf; Kaushik and Pathak, Pharm Biomed Res 2017;3(4):1-13. http://doi.org/10.18502/pbr.v3i4.84; etc.)
- Materials and Methods: In vitro release study: i) explain the purpose of adding SLS 1% w/v in the dissolution medium, ii) according to EMA and FDA guidelines for orally administered solid dosage forms, dissolution should be tested in the pH range of 1.2-6.8. Accordingly, the use of distilled water with 1% SLS w/v should be documented, iii) the performance of the prepared glimepiride tablet formulations should be also compared to that of a commercially available glimepiride tablet formulation in order to document the superiority of their proposed new formulations. In vivo study: i) crushing of the tablets before administration to the rats may mask the release characteristics of the tablets and lead to false conclusion about glimepiride absorption behavior, ii) explain the reason of using 1% CMC solution to suspend the crashed tablets, iii) the blood sample volume should be given. According to the NC3Rs Guidelines (https://www.nc3rs.org.uk/blood-sampling-general-principles) for repeated blood sampling from rats within 24 h, sample volume should be <1% of total blood volume, iv) the selection of the site of blood sampling should be documented. According to the NC3Rs Guidelines (https://www.nc3rs.org.uk/rat-retro-orbital-non-surgicalterminal), Retro-orbital bleeding should only be performed as a terminal procedure because of the severity of adverse effects that can occur with this technique, even in skilled hands. Furthermore, anaesthesia is required, v) The guidelines of Good Clinical Practice, the International Conference on Harmonization, and the European Medicines Agency, referred in page 5, 1st paragraph, should be added in the references list, vi) Give details on the type of PK analysis performed and the methodology used for the calculation of the PK parameters
- Results and Discussion: i) In Table 1, explain what "-" means for the disintegration time of 3D-printed tablets. Add explanation in the footnote of Table 1 ii) SEM image for the surface of the liquisolid tablet is not shown in Figure 2, iii) In Figure 3, symbols and profile is explained only for 3D-printed tablets, iv) Based on their in vitro release characteristics, 3D-printed and liquisolid tablets should not perform equally in vivo (as shown in Figure 3), since the 3D printed tablets show a controlled release behavior (reaching plateau at 6 h) while liquisolid tablets behave as immediate release formulation and reached plateau at 1h. Accordingly, crashing of tablets before administration to the rats, eliminated the controlled release behavior of 3D-printed SNEEDS based tablets and in the in vivo study the two SNEDDS based tablet (3D printed and liquisolid) formulations behaved in the same manner. Thus, the validity of the conclusion for the in vivo superiority of the 3D printed tablets is questionable and should be carefully reconsidered, v) Furthermore, the in vivo results should be discussed in comparison to the PK performance of the commercially available formulations.
- Conclusions: This section should be reconsidered and re-written after clarification and corrections made according to the above comments
- Explain each abbreviation the first time it is used (e.g., GLMP in page 4, first paragraph; PDI in page 5, 5th paragraph)
Author Response
General comments
The subject of the study falls within the scope of the Pharmaceuticals since it describes the preparation and characterization of liquisolid, directly compressed and 3D-printed glimepiride tablets, as well as their pharmacokinetic behavior evaluation.
However, the paper needs significant corrections, clarifications, and supplementation, as described below in the specific comments to the authors.
Specific Comments:
- Title: The title should emphasize on the tablet preparation and characterization and not on the short and questionable part (see below comments) of the PK study (e.g., Development of 3D-printed, liquisolid and directly compressed glimepiride tablets, loaded with black seed oil self-nanoemulsifying drug delivery system: in vitro and in vivo characterization)
Reply
Based on the reviewer suggestion, the title of the manuscript has been modified.
- Abstract: It should be corrected/re-written after corrections/clarifications and supplementation according to the comments below. Furthermore, the animal model used in the PK study should be specified in the abstract.
Reply
The abstract has been modified and the requested details has been added.
- Introduction: Glimepiride tablets are commercially available and in the SPC of the products is stated that the drug is rapidly and completely absorbed (100%) from the intestine after oral administration. The authors do not refer to the characteristics commercially available formulations in the introduction part, but they point that “the drug undergoes slow gastrointestinal dissolution and exhibits inter-personal variations, which affects its bioavailability”, giving only one old reference (ref 9: Badian et al, 1994). The authors should document their statement based on recent bibliography. In addition, the authors state that “Different strategies have been reported to enhance the dissolution and oral absorption of glimepiride, especially from tablets, such as solid dispersion and micronized techniques” giving only one relatively old research paper as reference (ref 11: Xiao Ning et al., 2011). However, a short search in literature revealed several papers dealing with the development of immediate or sustained release glimepiride tablet formulations (e.g., Qushawy et al., Sci. Pharm. 2020, 88, 52, https://doi.org/10.3390/scipharm88040052; Dhanorkar and Shah, Future J Pharm Sci (2021) 7:225, https://doi.org/10.1186/s43094-021-00374-5; Jogala et al., 2016, Int J Pharm Pharm Sci, Vol 8, Issue 5, 271-278; Gill et al., Asian Journal of Pharmaceutics - July-September 2010, http://doi.org/10.4103/0973-8398.72121; Glob J Pharmaceu Sci 1(2): Pagadala et al., GJPPS.MS.ID.55553, 2016; Siwach, et al., Nanoscience & Nanotechnology-Asia 10(5), 2020, http://doi.org/10.2174/2210681209666190422160115; Li et al., International Journal of Nanomedicine 2016:11 3777–3788, http://dx.doi.org/10.2147/IJN.S105419; Al-Madhagi et al., Journal of Pharmaceutics Volume 2017, Article ID 3690473, 5 pages https://doi.org/10.1155/2017/3690473; Arpita Pal et al., ACS Omega 2020, 5, 19968−19977, http://pubs.acs.org/journal/acsodf; Kaushik and Pathak, Pharm Biomed Res 2017;3(4):1-13. http://doi.org/10.18502/pbr.v3i4.84; etc.)
Reply
The introduction has been modified to include more recent references that mentioned different strategies to enhance the dissolution, absorption and bioavailability of glimepiride. Also, different glimepiride tablet formulations including immediate and modified release drug products have been added.
- Materials and Methods: In vitro release study: i) explain the purpose of adding SLS 1% w/v in the dissolution medium, ii) according to EMA and FDA guidelines for orally administered solid dosage forms, dissolution should be tested in the pH range of 1.2-6.8. Accordingly, the use of distilled water with 1% SLS w/v should be documented, iii) the performance of the prepared glimepiride tablet formulations should be also compared to that of a commercially available glimepiride tablet formulation in order to document the superiority of their proposed new formulations. In vivo study: i) crushing of the tablets before administration to the rats may mask the release characteristics of the tablets and lead to false conclusion about glimepiride absorption behavior, ii) explain the reason of using 1% CMC solution to suspend the crashed tablets, iii) the blood sample volume should be given. According to the NC3Rs Guidelines (https://www.nc3rs.org.uk/blood-sampling-general-principles) for repeated blood sampling from rats within 24 h, sample volume should be <1% of total blood volume, iv) the selection of the site of blood sampling should be documented. According to the NC3Rs Guidelines (https://www.nc3rs.org.uk/rat-retro-orbital-non-surgicalterminal), Retro-orbital bleeding should only be performed as a terminal procedure because of the severity of adverse effects that can occur with this technique, even in skilled hands. Furthermore, anaesthesia is required, v) The guidelines of Good Clinical Practice, the International Conference on Harmonization, and the European Medicines Agency, referred in page 5, 1st paragraph, should be added in the references list, vi) Give details on the type of PK analysis performed and the methodology used for the calculation of the PK parameters.
Reply
- Sodium lauryl sulphate was added as a commonly used surfactant in dissolution media for poorly water soluble drugs.
- The in vitro release of glimepiride from commercially available drug product has been added.
- We completely agree with the reviewer point of view about the fact that crushing of the tablets before animals administration may mask the release characteristics of the prepared tablets, but it was difficult to administer an intact tablet to the studied animals. Previous studies conducted the same procedure during administration of liquisolid and orodispersible tablets “references added”. A recommendation for clinical study on human volunteers has been mentioned in the revised manuscript to investigate the disintegration and release characteristics from an intact tablets.
- The blood volume and the requested details about blood sampling have been added to the revised manuscript.
- The requested refernces have been added.
- A non-compartmental pharmacokinetic model utilizing KineticaTM software (version 4, Thermo Electron Corporation, Waltham, MA, USA) was used to calculate the pharmacokinetics parameters. This information has been added to the revised manuscript.
- Pharmacokinetic parameters for the commercial available glimepiride tablets have been added to the revised manuscript.
- Results and Discussion: i) In Table 1, explain what "-" means for the disintegration time of 3D-printed tablets. Add explanation in the footnote of Table 1 ii) SEM image for the surface of the liquisolid tablet is not shown in Figure 2, iii) In Figure 3, symbols and profile is explained only for 3D-printed tablets, iv) Based on their in vitro release characteristics, 3D-printed and liquisolid tablets should not perform equally in vivo (as shown in Figure 3), since the 3D printed tablets show a controlled release behavior (reaching plateau at 6 h) while liquisolid tablets behave as immediate release formulation and reached plateau at 1h. Accordingly, crashing of tablets before administration to the rats, eliminated the controlled release behavior of 3D-printed SNEEDS based tablets and in the in vivo study the two SNEDDS based tablet (3D printed and liquisolid) formulations behaved in the same manner. Thus, the validity of the conclusion for the in vivo superiority of the 3D printed tablets is questionable and should be carefully reconsidered, v) Furthermore, the in vivo results should be discussed in comparison to the PK performance of the commercially available formulations.
Reply
- Detalis about the disintegration of 3D-printed tables has been added to table 1 and to the footnote of the same table.
- SEM images for the liquisolid tablets have been added.
- Figure 3 has been modified.
- A recommendation for clinical study on human volunteers using an intact tablets has been added to the revised manuscript to investigate the behavior of the 3D-printed, liquisolid and directly compresed tablet.
- Conclusions: This section should be reconsidered and re-written after clarification and corrections made according to the above comments.
Reply
The conclusion section has been modified.
- Explain each abbreviation the first time it is used (e.g., GLMP in page 4, first paragraph; PDI in page 5, 5th paragraph).
Reply
Abbreviations have been corrected.
Reviewer 3 Report
The aim of this study was to prepare the black seed oil based self-nanoemulsifying drug delivery system and its tablets. Furthermore, the authors a tablet was manufactured with 3D printer and compare the PK profiles using rats. However, the design of this research was not excellent, and the manuscript was also not well written. The reviewer wants to know some experimental procedures and results, but the authors did not mention them. For example, the physicochemical properties and the mechanism of this drug delivery system, etc. Therefore, the reviewer suggests that this manuscript needs extensive revision and adds the mechanism of this DDS in order to be published in Pharmaceuticals, and must be resubmitted.
- The title should be revised. “with black seed oil based self-nanoemulsifying drug delivery system”
- In abstract part, “0.273 mg/1g” should be checked.
- In introduction part, it is known that glimepiride is a very small drug loading with a dose of 2, 4, and 8 mg that needs to be taken. It is true that this drug is poorly soluble, but it is known that there is no problem with BA due to its small dose. According to reference 9, the BA has no problem when the tablets were administered. Thus, the authors should mentioned the novelty of this research and why SNEDDS needed.
- In accordance with Pharmaceuticals format, “Materials and Methods” should be section 3. The “Results and discussion” should section 2. “Results and discussion” should come before “Materials and methods”.
- The authors probably considered other fats before choosing black seed oil for the SNEDDS. It would be nice if that contents were added.
- Experimental data and results should support the hypothesis of this research. However, in this manuscript, it is judged that the proof of SNEDDS itself is insufficient.
- The background and purpose of using a 3d printer are also unclear.
- In Fig 1, why the authors used 0.1% SLS in dissolution media. In accordance with Pharmcopeia, 0.5% SLS was used. Why is the dissolution result of SNEDDS tablet and Liquisolid tablet divided into A and B? For comparison with in vivo, it is better to combine the figures
- Physico-chemical properties should add to understand the properties and mechanism of this SNEDDS
- In 3.5 in vivo section, the authors show only the results. Insightful interpretation of the results and a discussion of the reasons should be added.
Author Response
Comments and Suggestions for Authors
The aim of this study was to prepare the black seed oil based self-nanoemulsifying drug delivery system and its tablets. Furthermore, the authors a tablet was manufactured with 3D printer and compare the PK profiles using rats. However, the design of this research was not excellent, and the manuscript was also not well written. The reviewer wants to know some experimental procedures and results, but the authors did not mention them. For example, the physicochemical properties and the mechanism of this drug delivery system, etc. Therefore, the reviewer suggests that this manuscript needs extensive revision and adds the mechanism of this DDS in order to be published in Pharmaceuticals, and must be resubmitted.
- The title should be revised. “with black seed oil based self-nanoemulsifying drug delivery system”
Reply
The title of the manuscript has been revised.
- In abstract part, “0.273 mg/1g” should be checked.
Reply
The solubility of glimepiride in the prepared SNEDDS formulation was of 25 ± 0.273 mg/1g. This has been explained in the revised version.
- In introduction part, it is known that glimepiride is a very small drug loading with a dose of 2, 4, and 8 mg that needs to be taken. It is true that this drug is poorly soluble, but it is known that there is no problem with BA due to its small dose. According to reference 9, the BA has no problem when the tablets were administered. Thus, the authors should mentioned the novelty of this research and why SNEDDS needed.
Reply
To overcome glimepiride poor aqueous solubility, limited dissolution in the gastrointestinal fluids and to avoid dose dumping, we have developed a black seed oil based-SNEDDS formulation and used to prepare liquisolid and 3D-printed tablets. Directly compressed tablets containing pure glimepiride were also prepared and the in vitro and in vivo performances of the studied tablets were investigated. Black seed oil was incorporated in the SNEDDS formulation to improve glimepiride anti-diabetic activity.
This explanation has been added to the revised manuscript.
- In accordance with Pharmaceuticals format, “Materials and Methods” should be section 3. The “Results and discussion” should section 2. “Results and discussion” should come before “Materials and methods”.
Reply
The numbering for the “Results and discussion” and “Materials and Methods” have been revised.
- The authors probably considered other fats before choosing black seed oil for the SNEDDS. It would be nice if that contents were added.
Reply
All the studied oils have been mentioned in the revised manuscript.
- Experimental data and results should support the hypothesis of this research. However, in this manuscript, it is judged that the proof of SNEDDS itself is insufficient.
Reply
In this work, we aimed to improve glimepiride anti-diabetic activity by incorporation of black seed oil in the studied formulation and so SNEDDS was developed. Glimepiride showed higher solubility in the prepared SNEDDS formulation which was reflected in the in vitro release and pharmacokinetics.
This explanation has been added to the revised manuscript.
- The background and purpose of using a 3d printer are also unclear.
Reply
3D-printed tablets formulation was investigated in this work to shift from the conventional “one size fits all” formulation to personalized medicine as it is possible to change the drug concentration in the prepared formulation based on individual needs. The studied excipients in the 3D-printed tablets achieved extended drug release behavior which add extra benefits to this formulation.
This explanation has been added to the revised manuscript.
- In Fig 1, why the authors used 0.1% SLS in dissolution media. In accordance with Pharmcopeia, 0.5% SLS was used. Why is the dissolution result of SNEDDS tablet and Liquisolid tablet divided into A and B? For comparison with in vivo, it is better to combine the figures
Reply
The reason for using 0.1% SLS in the dissolution media has been clarified in the revised manuscript and supported with references.
We have divided the in vitro release into A and B as there was a difference in the release profile from the studied tablets. The 3-D printed tablets showed extended drug release up to 12 hours while the liquisolid exhibited maximum % drug release within 120 minutes. Accordingly, it will be difficult to merege all the formulation in one figure.
- Physico-chemical properties should add to understand the properties and mechanism of this SNEDDS
Reply
The prepared SNEDDS formulation was characterized by studting the size, polydispersity index and the drug load. Moreover, new image for the morphology of the prepared SNEDDS using the transmission electron microscopy has been added to the revised mabuscript.
- In 3.5 in vivo section, the authors show only the results. Insightful interpretation of the results and a discussion of the reasons should be added.
Reply
More details has been added to the revised manuscript.
Round 2
Reviewer 2 Report
General comments
The authors have made an effort to reply to my comments and revise their manuscript accordingly.
However, there are some points that need further revision.
Specific Comments:
- Abstract: The final conclusion that “The liquisolid and three-dimensional printed tables showed superior plasma profile and improvement in most of the calculated pharmacokinetic parameters. The relative bioavailabilities of the threedimensional printed and liquisolid tablets compared to directly compressed tablets were 187.36% and 181.68%, respectively. The liquisolid and three-dimensional printed tables are likely to become good choices and successful alternative to currently available directly compressed glimepiride tablets but clinical investigations are required”, is not correct and should be revised according to the comments below (see comment 3).
- Materials and Methods: In vivo study: Brief description of the methodology used for the calculation of the PK parameters should be given (e.g. calculation of terminal slope, half-life, AUMC, MRT, CL (or TCR), Vd), as well as the formula used for the calculation of relative bioavailability.
- Results and Discussion: i) The following previous comment is not replied by the authors, and not considered in the discussion of the results.
“Based on their in vitro release characteristics, 3D-printed and liquisolid tablets should not perform equally in vivo (as shown in Figure 3), since the 3D printed tablets show a controlled release behavior (reaching plateau at 6 h) while liquisolid tablets behave as immediate release formulation and reached plateau at 1h. Accordingly, crashing of tablets before administration to the rats, eliminated the controlled release behavior of 3D-printed SNEEDS based tablets and in the in vivo study the two SNEDDS based tablet (3D printed and liquisolid) formulations behaved in the same manner. Thus, the validity of the conclusion for the in vivo superiority of the 3D printed tablets is questionable and should be carefully reconsidered”
ii) The statement that liquisolid and three-dimensional printed tables showed superior plasma profile compared to the commercially available tablets (page 8, last paragraph) is not correct. Statistical comparison of the results would lead to statistically not significant differences between the PK parameters.
Furthermore, relative bioavailability should be calculated against the commercially available tablets.
iii) Taking into account the equal in vivo performance of the prepared SNEDDS based tablets with the commercially available formulation, the authors should carefully discuss and document the advantages of the developed glimepiride SNEDDS based formulations.
Conclusions: As for the abstract, this section should be reconsidered and re-written according to the above comments, and the comparison of the liquisolid and three-dimensional printed tables performance with that of the commercially available tablets.
Reviewer 3 Report
It is difficult to think that the comments of reviewers are properly addressed in the paper.
"25 ± 0.273 mg/1g", in accordance with international standards, the volume unit of g/L or mg/L(mL) should be applied when evaluating solubility. In addition, many errors are still seen, and it is considered difficult to publish in Pharmaceuticals.